# PC-YOLO11s: A Lightweight and Effective Feature Extraction Method for Small Target Image Detection

**DOI:** 10.3390/s25020348

**Published:** 2025-01-09

**Authors:** Zhou Wang, Yuting Su, Feng Kang, Lijin Wang, Yaohua Lin, Qingshou Wu, Huicheng Li, Zhiling Cai

**Affiliations:** 1College of Computer and Information Sciences, Fujian Agriculture and Forestry University, Fuzhou 350002, China; 52311049016@fafu.edu.cn (Z.W.); 52411049039@fafu.edu.cn (Y.S.); 52411049019@fafu.edu.cn (F.K.); 52411049022@fafu.edu.cn (H.L.); zhilingcai@126.com (Z.C.); 2Key Laboratory of Smart Agriculture and Forestry, Fujian Province University, Fuzhou 350002, China; 3School of Mathematics and Computer Science, Wuyi University, Wuyishan 354300, China; qswu@wuyiu.edu.cn

**Keywords:** small object detection, YOLO11, attention mechanism, VisDrone2019, tea bud

## Abstract

Compared with conventional targets, small objects often face challenges such as smaller size, lower resolution, weaker contrast, and more background interference, making their detection more difficult. To address this issue, this paper proposes an improved small object detection method based on the YOLO11 model—PC-YOLO11s. The core innovation of PC-YOLO11s lies in the optimization of the detection network structure, which includes the following aspects: Firstly, PC-YOLO11s has adjusted the hierarchical structure of the detection network and added a P2 layer specifically for small object detection. By extracting the feature information of small objects in the high-resolution stage of the image, the P2 layer helps the network better capture small objects. At the same time, in order to reduce unnecessary calculations and lower the complexity of the model, we removed the P5 layer. In addition, we have introduced the coordinate spatial attention mechanism, which can help the network more accurately obtain the spatial and positional features required for small targets, thereby further improving detection accuracy. In the VisDrone2019 datasets, experimental results show that PC-YOLO11s outperforms other existing YOLO-series models in overall performance. Compared with the baseline YOLO11s model, PC-YOLO11s mAP@0.5 increased from 39.5% to 43.8%, mAP@0.5:0.95 increased from 23.6% to 26.3%, and the parameter count decreased from 9.416M to 7.103M. Not only that, we also applied PC-YOLO11s to tea bud datasets, and experiments showed that its performance is superior to other YOLO-series models. Experiments have shown that PC-YOLO11s exhibits excellent performance in small object detection tasks, with strong accuracy improvement and good generalization ability, which can meet the needs of small object detection in practical applications.

## 1. Introduction

Small targets typically refer to objects that occupy a small area in an image and lack clear details, often making them difficult to detect due to low resolution and background interference [1]. Precise identification and localization through small target detection are of great significance. Small object detection has been an important research direction in the field of computers, and is also widely applied in many practical scenarios [1], such as the drone aerial and tea industries. In the drone aerial field, the drones can be applied in remote monitoring, target tracking, disaster detection, and so on, and typically fly at higher altitudes, resulting in smaller target sizes and complex backgrounds in the captured images. Additionally, in the field of agriculture, tea bud detection in tea production is also a very typical small object detection application, because tea bud detection has issues with color similarity to the background and small target size. Tea buds are the standard for picking. In intelligent agriculture, the precise identification of tea buds through small object detection for automated picking can help reduce labor costs and improve the quality of tea. Since the features are relatively blurry and have low contrast with the background, small target detection needs to overcome two problems. The first is the issue of size, as small targets may be obscured by the background or other objects, leading to a decrease in detection accuracy; The second issue is the lack of details, where the texture and shape information of small targets are difficult to capture and distinguish from complex backgrounds [2].

In the past, object detection mainly relied on traditional machine learning methods. However, they have significant limitations in small object detection. Firstly, traditional machine learning usually relies on manually designed features, which have a limited ability to capture the details of small targets, especially when the target size is small, as many details are easily lost. These methods are not suitable for dealing with complex backgrounds and multi-scale targets, and their training process is complex, lacking the adaptive ability of deep learning. Therefore, with the emergence of deep learning, it has become the mainstream research method for small object detection in recent years. Small object detection methods based on deep learning are mainly divided into two-stage and one-stage models. Among them, Faster-RCNN [3] is commonly used in two-stage models. Although two-stage detectors are relatively more accurate in detecting small targets, they are not suitable for mobile devices due to their high computational overhead and slow processing speed. So, currently, the most commonly used detectors are still one-stage detectors, mainly YOLO [4], SSD [5], Retina Net [6], etc. One-stage detectors can perform detection tasks at a high speed, and YOLO performs well in both speed and accuracy. However, the YOLO model generally suffers from low accuracy in small object detection tasks, susceptibility to background noise interference, and low resolution. This is because the convolution layers used in object detection often reduce the resolution of the image, especially in deeper convolution layers, which is very detrimental to the localization and detection of small targets [1]. So, the current research directions mainly include high-resolution image processing, multi-scale detection, network optimization, and data augmentation [7,8]. Additionally, some efforts have been made to propose improvement strategies for YOLOv3 [9], YOLOv5 [10], YOLOv8 [11], etc., to design YOLO models suitable for small object detection. With the proposal of the latest YOLO-series model, YOLO11 [12], this paper will use YOLO11s as the benchmark model to explore its performance in small target scenes. In order to enhance the ability of small target feature extraction and seek a balance between accuracy and parameter quantity, this paper will make an improvement to YOLO11, with the main contributions as follows:

(1)By adding a P2 layer in the network, finer features can be extracted in the high-resolution stage of the image, thereby improving the detection accuracy of small targets. At the same time, removing the P5 layer, which is mainly used for large object detection, can effectively reduce unnecessary calculations and decrease the parameters of the model.(2)The coordinate spatial attention mechanism module (CSA) was proposed and integrated, which was applied to the bottleneck module of C3k2, with C3k set to False, and the SPPF module. By enhancing the extraction of spatial and positional feature information, CSA helps to improve the detection performance of small targets and further enhance the detection accuracy of the model for small targets.(3)The experimental comparison results on the VisDrone2019 and tea bud datasets show that the PC-YOLO11 model outperforms other object detection models in terms of performance, especially providing important support for intelligent tea bud harvesting technology and a valuable reference for the real-time detection of small targets.

## 2. Related Works

### 2.1. Related Study Introduction

In order to better adapt to the unique features of small target images, researchers have proposed various innovative strategies to optimize existing detection models. Zhang et al. [13] introduced a bidirectional attention aggregation module and a TFD heuristic edge module. The former promotes the fusion of cross-horizontal features and enhances the shape representation ability of high-level features. The latter is used to extract effective edge features, helping to accurately predict the shape and mask of small targets, thereby enhancing the performance of small target detection. Yang et al. [14] designed a new query mechanism that first predicts the approximate location of small targets on low-resolution feature maps, and then uses sparse guidance to accurately calculate detection results using high-resolution features. This approach not only leverages the advantages of high-resolution feature maps but also avoids ineffective calculations on background regions. Chen et al. [15] combined Swin Transformer and a convolution module to enhance the model’s ability to capture global contextual information of small objects. At the same time, they also introduced a dual-level routing attention mechanism to enhance the attention to small targets, and through a dynamic detection head with deformable convolution and attention mechanism, the spatial perception ability of the model was enhanced, thereby improving the detection ability of small targets. Li et al. [16] added a detail enhancement module to the backbone network of YOLOv5 to better capture contextual information and improve target localization accuracy by combining attention mechanisms in the head. In addition, they also replaced the CIoU loss function with SIoU, thereby accelerating the convergence speed of the model. Wang et al. [17] proposed the TOCM method to improve the difference between the background and target, thereby expanding the detection boundary of the model for the background and target. They also designed a fine-grained feature fusion module (SFM) to improve the model’s ability to express semantic features at different scales, thereby enhancing the detection accuracy of small targets. Ma et al. [18] improved the model’s perception ability of multi-scale targets based on YOLOv8 and combined it with a Bidirectional Feature Pyramid Network (BiFPN). They also replaced the traditional cross-row convolution layer with a spatial to depth layer to enhance the extraction of fine-grained information and small-scale target features. In addition, they also adopted the Multi-SEAM module to enhance the learning of occluded target area features, and finally improved the accuracy and robustness of the model in small object detection through four detection heads. These improvement methods not only improve the accuracy of small object detection, but also bring an increase in the number of parameters. In order to meet the performance requirements of unmanned aerial vehicles with limited computing power in aerial target detection, Xu et al. [19] improved the YOLOv8n model. They introduced multi-scale fusion technology and used conditional parameterized convolution to enhance the feature representation ability of the model. In addition, in order to improve the accuracy of bounding box regression, the Wise IoU loss function was introduced to more effectively optimize the prediction of bounding boxes. Shi et al. [20] mainly studied a small object detection method based on Focus-Det. This method uses STCF-EANet as the backbone of the feature extraction network to more effectively extract image features. Meanwhile, utilizing the Bottom Focus PAN module to capture a wider range of small object positions and low-level detail information. In order to solve the common problem of missed detection in dense small object scenes, SIoU-SoftNMS was also introduced to optimize the detection performance. With the continuous advancement of computer vision and deep learning technologies, the application prospects of small object detection will become even broader. Currently, there has been research on the application and improvement of the YOLO11 model. For instance, Li et al. [21] introduced an efficient channel attention mechanism to enhance the model’s ability to focus on significant features, thereby improving detection accuracy. Additionally, they modified the loss function to increase the model’s attention to uncertain target areas, further enhancing its robustness. This approach was successfully applied to a dataset for underground coal miner detection. It is anticipated that the application and improvement of the YOLO11 model will become increasingly diverse and advanced in the future.

### 2.2. YOLO11 Object Detection Algorithm

YOLO11 is the latest version of the Ultralytics YOLO series, which demonstrates outstanding accuracy, speed, and efficiency in real-time object detection, redefining the potential of computer vision technology. Based on significant advancements in previous generations of YOLO versions, YOLO11 has undergone significant optimizations in network architecture and training methods, making it an ideal choice for handling various computer vision tasks. In terms of model structure, YOLO11 has made multiple improvements to enhance feature extraction capabilities and overall model efficiency. Firstly, it replaces the original C2f module with a C3k2 module, enhancing the flexibility of the module in different application scenarios. Next, a C2PSA module was added after the SPPF module, which enhanced the model’s attention mechanism in feature extraction by extending the C2f structure and introducing PSA, thereby improving its ability to capture key features. Specifically, the C2PSA module adopts a multi-head attention mechanism and a feedforward neural network (FFN) to further optimize the feature extraction process. The multi-head attention mechanism enables the model to focus on multiple dimensions of input features, while FFN helps the model map input features to a higher dimensional space, thereby capturing more complex nonlinear relationships and improving feature representation capabilities. At the same time, the C2PSA module can also selectively add residual connections (shortcuts), which helps improve gradient propagation and accelerate network training. Finally, YOLO11 uses depth-wise separable convolution in detecting the head, which not only reduces redundant calculations but also improves the computational efficiency of the model. Overall, YOLO11 adopts an improved backbone network and neck architecture, enhancing feature extraction capabilities and significantly improving the effectiveness of feature extraction, thereby improving the accuracy of object detection. Moreover, through streamlined architecture design and optimized training processes, YOLO11 achieves faster processing speeds while maintaining efficient performance and ensuring accuracy. Through these design optimizations, YOLO11 has achieved a better balance between performance and efficiency. The YOLO11 model structure is shown in Figure 1 and Figure 2. Although significant progress has been made in the field of object detection, there are still some issues, including poor performance in detecting small objects. As the depth of the model increases, it can lead to the loss of small target features or interference from background noise. Therefore, ensuring that the network effectively obtains feature information of small targets is a challenge that needs to be addressed.

## 3. PC-YOLO11 Object Detection Algorithm

### 3.1. Optimization of Detection Layer Structure

We optimized the YOLO11 network structure by adding the P2 detection layer while removing the P5 detection layer. The core goal of this adjustment is to enhance the detection capability of small targets and reduce the complexity and parameters of the model. Specifically, the P2 detection layer is located at a lower level of the network and can better capture detailed features, which is particularly important for detecting small targets. In practical applications, many scenarios require high-precision small object detection, such as pedestrians, animals, or small vehicles in surveillance videos. By introducing the P2 detection layer, YOLO11 can more accurately identify these small targets, effectively reducing missed detection. On the other hand, the P5 detection layer is located at a higher level of the network and typically extracts abstract features that are often affected by background noise and have limited assistance in detecting small targets. Therefore, after removing the P5 layer, YOLO11 can focus more on feature extraction in the middle and lower layers, improve the recognition efficiency of small targets, and make the network more efficient and concise in processing. Since the P2 detection layer is located in the shallower layers of the network, its feature maps have higher resolution but fewer channels, resulting in relatively lower parameter requirements. In contrast, the P5 detection layer, often positioned in deeper network layers, incurs higher parameter costs to process more complex and abstract features.

The P2 detection layer is located in a lower layer of the YOLO11 network and is typically used to capture finer-grained feature information. In object detection tasks, small targets often have lower resolution and less detail information, so the network needs to learn and pay more attention to these details in low-level feature extraction. Traditional object detection methods often focus on abstracting high-level features, but for small targets, these high-level abstract features often lack sufficient details, resulting in poor recognition performance. As a low-level feature extractor, the P2 layer can better capture the detailed information of small targets because it directly acts on high-resolution feature maps and can more accurately preserve local details of the target, such as edges, textures, and shapes, which are crucial in small target detection. By introducing the P2 layer, YOLO11 can learn the key features of small targets at lower network levels, significantly improving the detection accuracy of small targets and avoiding missed detection.

Compared to the P2 layer, the P5 layer is located at a higher layer of the YOLO11 network and is typically used to extract more abstract features. These high-level features undergo multiple convolutions and pooling processes, usually focusing more on the global information of the image, such as large-scale objects and the overall background. However, for small object detection, the high-level features of the P5 layer may not be sensitive to capturing detailed information and are easily affected by background noise interference. Therefore, in small object detection tasks, the role of the P5 layer is limited and may even introduce redundant information, affecting the recognition efficiency of the model. After removing the P5 detection layer, the network structure of YOLO11 is simplified, with a focus on extracting low-level and mid-level features, allowing the network to focus more on detecting small targets. In these layers, the resolution of the feature map is higher, which can preserve more detailed information, especially for small objects. The features in the lower and middle layers usually provide better recognition. In addition, removing the P5 layer effectively reduces the number of parameters and computational overhead of the model, and avoids the possibility of excessive dependence on high-level features, making the model more flexible, stable, and more suitable for small object detection needs in practical applications. The improved model structure is shown in Figure 3.

### 3.2. Coordinate Spatial Attention Mechanism

To further enhance the feature extraction of small targets in images, we have designed a spatial and coordinate hybrid attention mechanism. Firstly, small targets usually occupy fewer pixels in the image and are easily overlooked. Through this attention mechanism, it is possible to effectively focus on small target areas in the image, enhance the model’s attention to these features, and thus improve the recognition rate of small targets. Secondly, small targets are often disrupted by complex backgrounds. This attention mechanism can first integrate spatial feature information and then utilize coordinate information to enhance the modeling ability of local context, helping the model separate useful information when processing small coordinates, thereby reducing the influence of background noise. Thirdly, in small object detection, features of different scales are crucial for accurate localization. Therefore, we have added attention mechanisms to the SPPF [22] module, which can establish effective connections between features at different levels, enabling better fusion of small target features with global features and improving detection accuracy. Fourthly, small targets have different positions and shapes in different images. The hybrid attention mechanism can dynamically adjust attention weights based on coordinates, making the model more flexible in adapting to changes in small targets in each sample. Finally, the diversity and complexity of small targets may lead to difficulties in training the model. Introducing spatial and coordinate information can help the model better understand the relative position and shape features of the targets, thereby enhancing its robustness to small targets. The specific implementation of this mixed attention is shown in Figure 4.

For small targets, spatial information is crucial, and spatial attention maps are generated by utilizing the spatial relationships between features. Unlike channel attention, spatial attention focuses on the “where” part of the information. To calculate spatial attention, we used SAM [23], which first applies average pooling and max pooling operations along the channel dimension and concatenates them to generate effective feature descriptors. Then, weight features with spatial information are obtained through convolution and activation functions, and finally multiplied with the original feature map to obtain a feature map with spatial information. To further extract the feature information of small targets, we further obtain positional information on the feature map of spatial information. Firstly, since small targets are easily disturbed by background noise, we perform max pooling operations along the X and Y directions, respectively [24]. Max pooling selects the maximum value in each pooling window, which helps preserve the salient features in the image, is more sensitive to the edge and texture information of small targets, effectively suppresses background noise, and helps improve the recognizability of small targets. Meanwhile, max pooling can provide translation invariance to a certain extent, enhancing the robustness of the model to small target position changes. Afterwards, the two feature maps are concatenated, and then a 1 × 1 convolution is used for dimensional reduction and activation to generate a new feature map. Using this feature map, a split operation is performed along the spatial dimension. Finally, a 1x1 convolution is used for dimensional enhancement and sigmoid activation function to obtain the final attention weight. This weight is point multiplied by the feature map with spatial information to obtain the final feature map with spatial and positional information.

In this study, we propose a novel module called CSA_Bottleneck that combines an attention mechanism with a bottleneck structure. The design of this module aims to effectively extract feature information of small objects to improve the recognition ability of object detection models for small objects. Specifically, when C3k is set to False, the CSA_Bottleneck module can replace the traditional bottleneck structure by introducing attention mechanisms in shallower network layers, thereby better focusing on the features of small targets and improving detection accuracy.

The core idea of the CSA_Bottleneck module is to combine an attention mechanism with a bottleneck structure, which can dynamically adjust the attention area of the network during the feature extraction process. The bottleneck structure can effectively reduce computational complexity by compressing and re-expanding feature channels. However, when extracting small objects, traditional bottleneck structures may not be able to effectively capture key information about small objects due to the loss of local details. The introduction of the attention mechanism can guide the network to focus more on the area where small targets are located when extracting features, thereby avoiding this information loss. We found in the experiment that when C3k is set to False, this combination can effectively improve the performance of the model in small object detection tasks.

However, when C3k is set to True, due to the deep network hierarchy, the CSA_Bottleneck module with an attention mechanism may actually cause information redundancy, leading to a decrease in model accuracy. Deep networks are already able to abstract higher-level feature information, and further adding attention mechanisms may overly focus on certain regions, resulting in wasted computing resources and easily introducing redundant attention weighting. Therefore, the CSA_Bottleneck module only has its maximum utility in shallower network layers.

In addition to integrating with bottleneck structures, we also combined the CSA attention mechanism with the Spatial Pyramid Pooling (SPPF) module and placed it after the first convolution layer of SPPF. The SPPF module can capture rich contextual information through pooling operations at different scales, thereby enhancing the model’s ability to handle scale changes, especially in multi-scale object detection tasks, where the SPPF module has significant advantages. SPPF helps the model maintain high accuracy when facing targets of different sizes by fusing contextual information at multiple scales. However, due to the challenges brought by scale changes, it is crucial to effectively capture the features of small and large targets. In order to further improve the detection accuracy of small targets, we placed the CSA attention mechanism at the front end of the SPPF module to ensure that the model can more accurately identify the key information of small targets before performing multi-scale fusion. In this way, the attention mechanism can guide the network to focus on small target areas, which helps to obtain more accurate features before scale fusion, thereby improving the final detection accuracy. The improved modules are shown in Figure 5.

Through this modular design, the model exhibits stronger robustness when dealing with scale changes. The SPPF module integrates multi-scale information to enable the model to find a suitable balance between targets of different sizes, thereby improving the adaptability of the model to targets of different scales. The CSA_Bottleneck module, which combines attention mechanisms, further enhances the sensitivity of the model to small target features. In addition, this improvement also helps to solve common problems in small object detection, such as loss of target details and low contrast. Guided by attention mechanisms, the network can automatically focus on key features of small targets, while CSA_SPPF further enhances overall robustness through multi-scale fusion of contextual information, enabling the model to exhibit high stability and accuracy when facing different types of targets.

The improvement of the CSA_Bottleneck module and CSA_SPPF module combined with an attention mechanism can effectively enhance the performance of the model in small object detection tasks. By introducing attention mechanisms at specific levels, we can optimize the feature extraction process and better address the challenge of detecting small targets. At the same time, the multi-scale information fusion of the SPPF module improves the adaptability of the model to targets of different sizes, thereby enhancing the robustness of the model in practical applications. Ultimately, through this design, the model is able to improve accuracy while avoiding redundant information issues caused by deep networks, achieving a balance between performance and efficiency. The specific effects will be demonstrated in Section 4. The improved YOLO11 model structure is shown in Figure 6.

## 4. Experiment

Next, we will validate the performance of the PC-YOLO11s model. Firstly, we will evaluate PC-YOLO11s using the VisDrone2019 datasets [25] and validate the effectiveness of the model improvement through ablation experiments and five different experiments, including applying the improvement strategy to other models, comparing PC-YOLO11s with other YOLO-series models, analyzing the effectiveness of the detection layer, evaluating the role of attention mechanisms, and comparing the effects of adding CSA at different positions. Finally, we will also apply PC-YOLO11s to tea bud datasets [26] and conduct comparative experiments with other YOLO-series models.

### 4.1. Datasets

This study mainly uses the VisDrone2019 datasets to evaluate our model, which are comprehensive datasets for drone vision research, mainly used for tasks such as object detection and tracking. The datasets contain a large number of images captured by drones, with a total of 8629 images, including 6471 for training, 548 for validation, and 1610 for testing. It covers different scenarios and targets, including ten categories such as pedestrians, vehicles, and bicycles. Most images have high resolution and can provide clear target details and scene information. The partial images and prediction results of VisDrone2019 are shown in Figure 7.

### 4.2. Experimental Environment

All experiments in this study were based on the Pytorch deep learning framework and programmed in Python 3.10. Experimental platform hardware environment: The CPU is 18 vCPU AMD EPYC 9754 128-Core Processor, the GPU is NVIDIA GeForce RTX 4090D 24G, the memory is 80GB, and the operating system is 64-bit Windows. The specific parameter settings are as follows: learning rate of 0.001, momentum of 0.937, weight decay factor of 0.0005, batch size of 8, number of iterations of 300, and image size of 640 × 640.

### 4.3. Model Evaluation Metrics

This article evaluates the algorithm using average precision mean (*mAP*), precision (*P*), recall (*R*), *F*1, and parameter count as evaluation metrics.

*mAP* is used to measure its accuracy across various categories, and its formula can be expressed as(1)$$AP_{i}=\int_{0}^{1}P_{i}\left(R\right)dR$$(2)$$mAP=\frac{\Sigma_{1}^{N}AP_{i}}{N}$$

*N* represents the number of categories.

Accuracy refers to the proportion of true targets detected by the model among all positive targets (i.e., all boxes predicted as targets), and its formula can be expressed as(3)$$P=\frac{TP}{TP+FP}$$

The recall rate refers to the true targets that the model can detect (i.e., the proportion of all true targets detected), which measures the comprehensiveness of the targets detected by the model relative to all actual targets. Its formula can be expressed as(4)$$R=\frac{TP}{TP+FN}$$

TP is true positive, indicating the actual number of positive classes predicted as positive; FP is false positive, indicating the actual number of negative classes predicted as positive; FN is false negative, indicating the actual number of positive classes predicted as negative.

The *F*1 score is the harmonic mean of precision and recall, used to balance the impact between the two. When two models are not equally divided in their evaluation metrics, the *F*1 score can be used for further evaluation, and its formula can be expressed as(5)$$F1=2\times \frac{Precision\times Recall}{Precision+Recall}$$

### 4.4. Comparative Analysis with Other YOLO-Series Models

To validate the effectiveness of the proposed model improvements, we applied the modifications to YOLO11 models of different scales, including YOLO11n, YOLO11s, and YOLO11m. The improved models were named PC-YOLO11n, PC-YOLO11s, and PC-YOLO11m, respectively. Table 1 provides a detailed comparison of the experimental results for each model, demonstrating the performance enhancement achieved through the improvements.

As shown in Table 1, the improvement strategies are equally applicable to other YOLO11 models. Moreover, as the model size increases, accuracy also improves. Notably, all the improved models demonstrate higher accuracy compared to their original counterparts. However, to achieve a balance between accuracy and model complexity, we selected PC-YOLO11s as the final model proposed in this study.

To further show the effectiveness of the proposed improvement strategies, P234 and CSA were integrated into YOLOv5 and YOLOv8, which have structural similarity to YOLO11 and were then named PC-YOLOv5 and PC-YOLOv8, respectively. The relevant comparative experimental results are shown in Table 2.

From the experimental results in Table 2, it can be seen that this improvement strategy has also achieved significant performance improvement on YOLOv5 and YOLOv8, verifying the wide applicability of the method. Although PC-YOLOv8s performs better in various indicators, its performance improvement is not very significant compared to PC-YOLO11s, and it has a larger number of parameters. Therefore, when balancing accuracy and parameter quantity, we believe that PC-YOLO11s can better control the complexity and computational cost of the model while maintaining high accuracy. Therefore, we ultimately chose PC-YOLO11s as our final model to achieve the best balance between accuracy and efficiency.

To further show the advantage of PC-YOLO11s, we compared our model with other mainstream models, including YOLOv5s, YOLOv6s [27], YOLOv7 tiny [28], YOLOv8s, YOLOv9s [29], YOLOv10s [30], YOLO11s, and PC-YOLO11s. The experimental results are shown in Table 3.

From the results in Table 3, it can be seen that our proposed model has better performance and lower parameter count. Although YOLOv5s and YOLOv7-tiny have lower parameter counts, their performance in accuracy and other aspects is much worse. YOLOv6s, YOLOv8s, YOLOv9s, and YOLOv10s have high detection accuracy, but they are still 2% to 5% worse than our model, and they also have higher parameter counts. Compared with the baseline model YOLO11s, it improved accuracy by 4.1%, recall by 3.3%, mAP@50 by 4.3%, mAP@50:95 by 2.7%, and F1 score by 3.7%, and the number of parameters decreased to 75% of the original. The experimental results once again demonstrate the effectiveness of our model in small object detection.

### 4.5. Ablation Analysis

To evaluate the contribution of P234 and CSA to the overall performance of PC-YOLO11s, we conducted ablation experiments on the VisDrone2019 datasets. The experimental design includes four different model configurations, namely baseline model, modified detection layer, model with added attention mechanism, and final improved model. The experimental results are summarized in Table 4, where “√” means the corresponding strategy is used and “×” means the corresponding strategy is not used.

According to the experimental results, adding the P2 detection layer and removing the P5 detection layer significantly improved the performance of the model, with a 3.4 percentage point increase in mAP@50 value. This change indicates that lower levels of the network, such as the P2 layer, contain richer feature information about small targets. Traditional object detection networks typically handle larger targets at higher levels (such as P5 layers), but these layers often lose some detailed features about small targets. Therefore, by introducing additional detection layers in the lower layers of the network, the model can better capture the local details and positional features of small targets, thereby improving the recognition accuracy of small targets. In addition, by removing the P5 detection layer, the parameter count of the model was reduced by 24.7%. This not only optimizes the storage and computational efficiency of the model, but also indicates that excessively deep network layers may not bring significant performance improvements in small object detection, but instead increase redundant calculations and parameters. Through this optimization, the computational complexity of the model has been effectively reduced, while maintaining good accuracy performance in small object detection tasks, thus verifying the effectiveness of this network structure adjustment. After introducing an attention mechanism, although applying an attention mechanism alone did not significantly improve the performance of the model, when combined with the above improvements (such as adding a P2 detection layer and removing the P5 layer), the mAP@50 value increased by 4.3 percentage points compared to the baseline model, and the number of parameters was only 75% of the original. This indicates that the attention mechanism can more effectively guide the network to focus on important feature regions related to small targets in the image, thereby improving detection performance. By weighing the spatial and channel information of the input image, the attention mechanism enables the network to focus more on key and recognizable small target areas, thereby improving the overall detection accuracy. The effect of this optimization is not only reflected in the improvement of performance, but also in the improvement of computational efficiency. After introducing the attention mechanism, although the complexity and parameter count of the model have increased, its computational cost is still lower compared to traditional models. Specifically, the attention mechanism plays an important role in enhancing the sensitivity of the model to small targets while avoiding excessive redundant calculations, thus maintaining high computational efficiency while ensuring accuracy. Overall, by adding the P2 detection layer, removing the P5 detection layer, and introducing an attention mechanism into the network, the overall performance of the model in small object detection tasks has been significantly improved. This optimized combination not only enhances the model’s ability to recognize small targets but also effectively reduces computational and storage overhead, providing strong support for efficient small target detection in practical applications. To more intuitively verify the effectiveness of the improved method, we compared the above four experiments through images, as shown in Figure 8.

### 4.6. Analysis of the Effectiveness of Different Detection Layers

From the data in Table 3, it can be seen that after improving the detection layer, the experimental results have been significantly improved. Next, we will analyze and compare the settings of different detection layers. Specific experiments were conducted using P345 (i.e., original YOLO11), P2345, P34, and P234, and the experimental results are shown in Table 5.

From the experimental results, it can be seen that any setting that includes a P2 detection layer can significantly improve accuracy. Especially when P2 and P5 coexist, the accuracy performance is the most superior, but at the same time, the number of parameters in the model is also relatively large. In order to find a balance between accuracy and parameter quantity, we chose P34 as the benchmark to further compare and analyze the experimental effects of P345 and P234. When P34 changes to P345, although mAP@50 only increases by 0.4% and mAP@50:95 only increases by 0.3%, the parameter count increases by 37%. When P34 changed to P234, mAP@50 increased by 3.4%, mAP@50:95 also increased by 2.1%, and the parameter count only increased by 2%. Based on this analysis, we ultimately chose P234 as the first experimental improvement plan, in order to control the increase in parameter quantity while ensuring accuracy improvement.

### 4.7. Analysis of the Effectiveness of the Attention Mechanism

To verify the effectiveness of the coordinate space attention mechanism proposed in this article, we conducted comparative experiments by adding a P2 detection layer and removing a P5 detection layer, and introducing CBAM [23], CA [24], and CSA at the same position. The experimental results are shown in Table 6.

From the experimental results, it can be seen that regardless of which attention mechanism is chosen, the accuracy of the model has been significantly improved, indicating that introducing attention mechanisms in the detection process can effectively enhance the performance of the model. In particular, our coordinate space attention mechanism (CSA) is 0.6% higher on mAP@50 compared to CBAM. This result indicates that for small target detection tasks, spatial and positional features are more critical in information expression than spatial and channel features. This may be related to the local features and spatial distribution of small targets, so it is possible to better capture the details of small targets through position-aware mechanisms. In addition, although there is not much difference in overall accuracy between CA and CSA, CSA outperforms CA in the harmonic mean of F1 scores. The F1 score, as a comprehensive evaluation indicator of accuracy and recall, can effectively measure the performance of the model when dealing with imbalanced datasets. Therefore, CSA not only improves the accuracy of the model but also optimizes its overall performance in small object detection tasks. Further analysis indicates that simple channel attention or spatial attention can capture certain key information when dealing with small targets. However, CSA can more accurately enhance the identification ability of small target areas by comprehensively considering spatial and positional features. Especially in situations where the target size is small and the background is complex, CSA can effectively improve the model’s ability to locate and recognize small targets, reducing false positives and false negatives. Therefore, based on these experimental results, we can conclude that the coordinate space attention mechanism can not only improve the accuracy of small object detection but also significantly improve the generalization ability of the model, making this mechanism widely applicable in more complex detection tasks.

Not only that, we also designed comparative experiments on the location of attention addition, which were performed only on the C3k2 module with C3k = False and on all C3k2 modules (including C3k = False and C3k = True). The experimental results are shown in Table 7.

From the experimental results, it can be seen that when the CSA attention mechanism is added to all C3k2 modules, the performance of the model actually decreases, especially in mAP@50, which is 0.2% lower than that after adding the attention mechanism only to C3k2 modules when C3k = False. This result indicates that the depth of the C3k2 module is deeper when C3k = True, which poses certain challenges for the network in extracting features of small targets. Although deep networks can extract more abstract high-level features, they often lose local details when processing small targets, resulting in a decrease in the recognition accuracy of small targets. Especially in deep networks, the process of information transmission is prone to gradient vanishing or feature blurring, which weakens the network’s ability to capture features of small targets. In addition, the purpose of incorporating the CSA attention mechanism is to enhance the sensitivity of the model to small targets by guiding the network to pay more attention to important feature regions. However, in deep networks with C3k = True, due to the feature maps already being at a high level of abstraction, attention mechanisms may introduce redundant attention regions, which in turn increases the computational burden of the network and does not effectively improve the detection performance of small targets. Redundant attention weighting not only fails to enhance the model’s recognition of small targets but may also lead to excessive attention to irrelevant regions, thereby affecting the improvement of accuracy. Therefore, based on this experimental result, this paper chooses to add the CSA attention mechanism only to the C3k2 module when C3k = False. Through this refined module selection, the improvement in this article achieves a balance between performance and efficiency, which can improve the accuracy of small object detection while avoiding performance degradation caused by too-deep network layers and redundant attention.

### 4.8. Application in Tea Bud Datasets

To validate the effectiveness of our model, we will continue by using tea bud datasets for testing. The comparison models used include YOLOv5s, YOLOv6s, YOLOv7-tiny, YOLOv8s, YOLOv9s, YOLOv10s, YOLO11s, and PC-YOLO11s. We cropped the original images in the tea bud datasets on the kaggle platform and processed them with brightness enhancement, dimming, blurring, flipping, and other image processing techniques. A total of 6250 images were obtained, which were divided into a training set, testing set, and validation set in an 8:1:1 ratio. Finally, we obtained 5000 images in the training set, 625 images in the validation set, and 625 images in the testing set. Original tea bud images and prediction results are shown in Figure 9. The tea buds in the tea bud datasets have the problem of small target size and similar target color to the background, which is very similar to the problem we are trying to solve. We set the iteration number for this experiment to 200 and the batch size to 16. The specific experimental comparison results are shown in Table 8.

From Table 8, it can be seen that our proposed model still performs better than other mainstream models and baseline models in terms of accuracy, recall, and precision in the tea bud datasets, and has significant advantages in parameter quantity, meeting the flexible deployment needs of mobile devices. Compared with the baseline model, our proposed model has improved accuracy by 0.7%, recall by 1.4%, mAP@50 by 0.4%, and mAP@50:95 by 2.1%, and reduced parameter quantity to 75.4% of the original. The experimental results further demonstrate the effectiveness and robustness of our model in small object detection.

## 5. Conclusions

In response to the poor performance of YOLO11 in small object detection tasks, this paper proposes the PC-YOLO11s model to solve this problem. There are two main improvements to this model. The first is the enhancement of the ability to extract small target features and reduce unnecessary computation and model complexity by adding a P2 detection layer and removing a P5 detection layer. The second is the proposal of a CSA attention mechanism and its integration into the network module to preserve more detailed small target feature information. The experimental results show that PC-YOLO11s outperforms other YOLO-series models on the VisDrone2019 and tea bud datasets. Compared to YOLO11s, mAP@50 increased by 4.3% and 0.4%, respectively, and mAP@50:95 increased by 2.7% and 2.1%, respectively. PC-YOLO11s has demonstrated excellent accuracy and good generalization ability in small object detection tasks, and has strong practical application potential. In the future, we will continue to optimize the network structure and explore modules for model pruning and effective feature extraction, aiming to enhance the deployment and application of the model in situations where computing resources are limited.

## Figures and Tables

**Figure 1 sensors-25-00348-f001:**
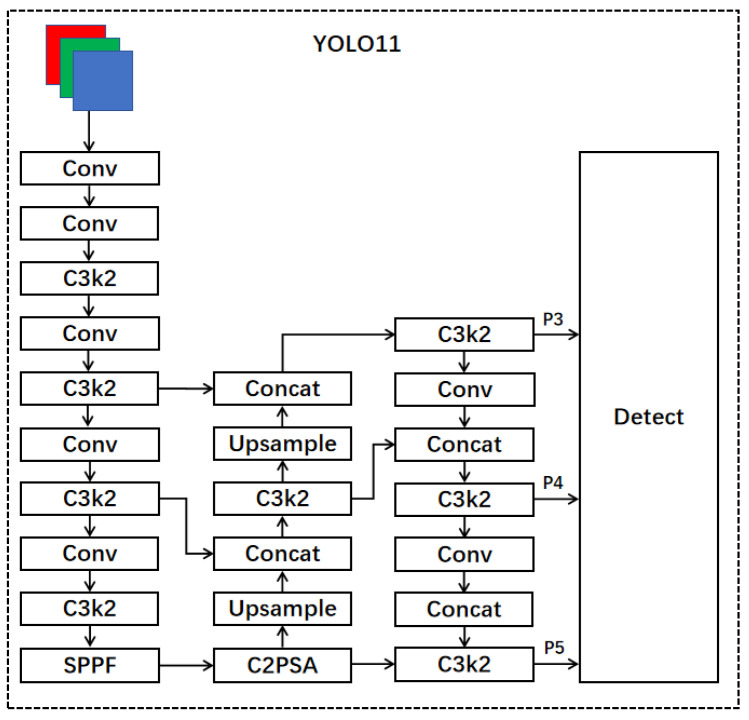
The network structure of YOLO11.

**Figure 2 sensors-25-00348-f002:**
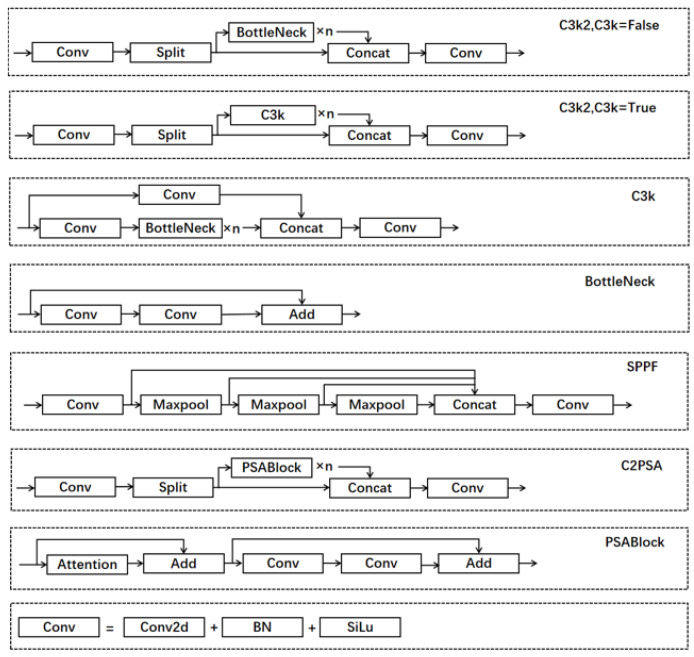
The network structure of some modules.

**Figure 3 sensors-25-00348-f003:**
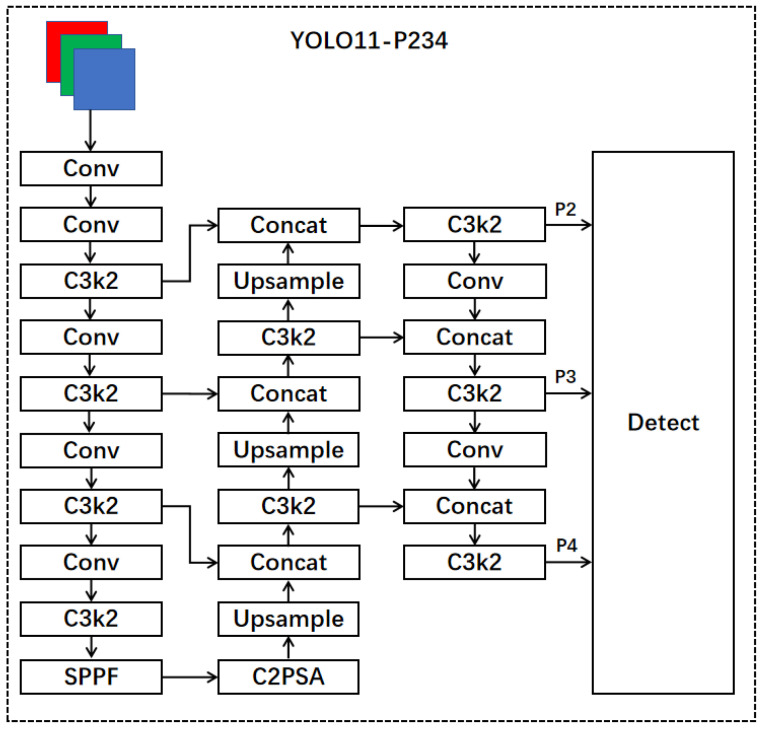
The model structure after modifying the detection layer.

**Figure 4 sensors-25-00348-f004:**
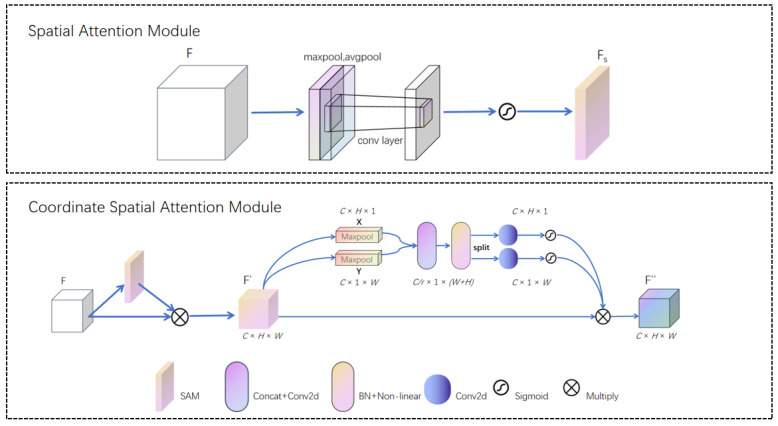
Structure of SA and CSA modules.

**Figure 5 sensors-25-00348-f005:**
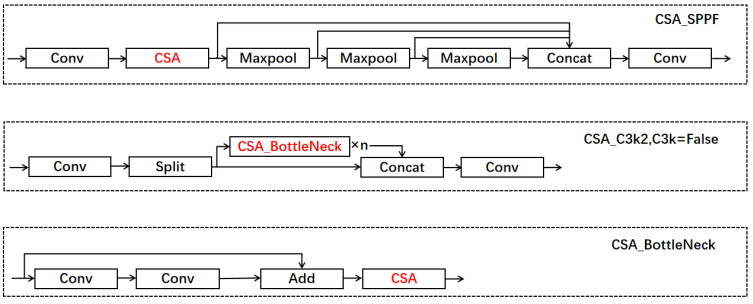
The network structure of some modules.

**Figure 6 sensors-25-00348-f006:**
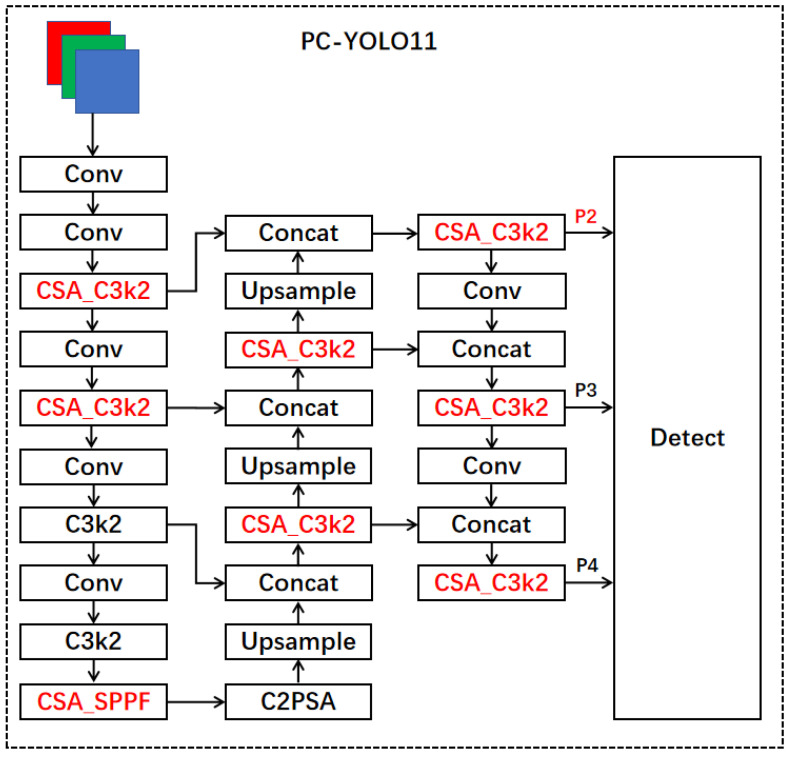
The network structure of PC-YOLO11.

**Figure 7 sensors-25-00348-f007:**
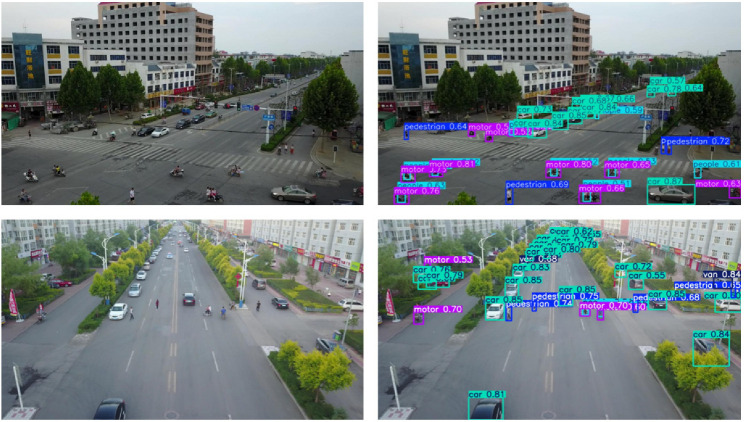
Visualization of partial images and their detection results in the VisDrone2019 datasets.

**Figure 8 sensors-25-00348-f008:**
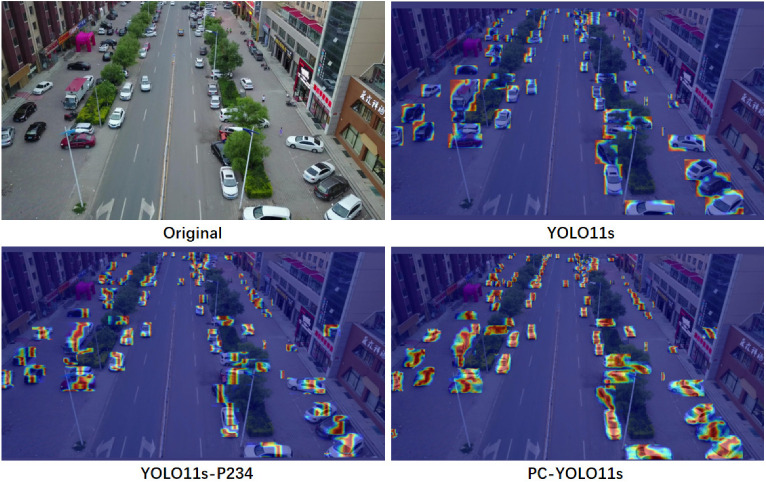
Comparison of heat map and original images of detection results in various stages of ablation experiment.

**Figure 9 sensors-25-00348-f009:**
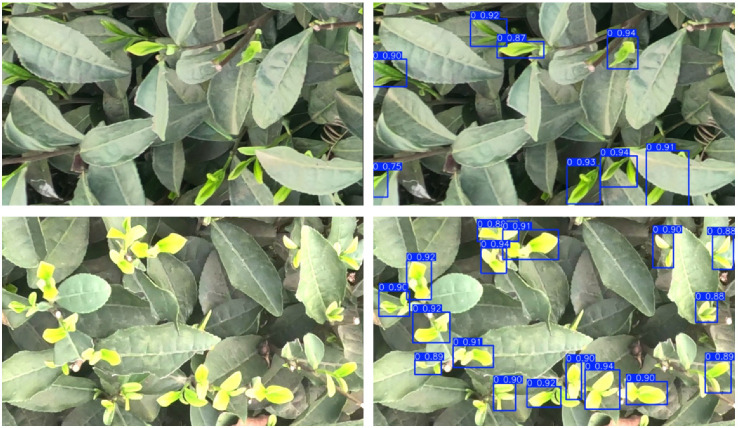
Visualization of original images and their detection effects in the tea bud datasets.

**Table 1 sensors-25-00348-t001:** Comparison of improvement strategies in YOLO11n, YOLO11s, and YOLO11m.

Model	P/%	R/%	mAP@0.5/%	mAP@0.5:0.95/%	F1	Parameter/M
YOLO11n	44.5	33.6	33.4	19.5	38.3	2.584
PC-YOLO11n	46.8	35.4	36.1	21.5	40.3	1.944
YOLO11s	50.2	38.6	39.5	23.6	43.6	9.416
PC-YOLO11s	54.3	41.9	43.8	26.3	47.3	7.103
YOLO11m	53.5	43.3	44.0	26.9	47.9	20.037
PC-YOLO11m	57.2	46.7	48.6	30.0	51.4	16.111

**Table 2 sensors-25-00348-t002:** Comparison of improvement strategies in YOLOv5s, YOLOv8s, and YOLO11s.

Model	P/%	R/%	mAP@0.5/%	mAP@0.5:0.95/%	F1	Parameter/M
YOLOv5s	46.3	33.4	33.4	18.1	38.8	7.037
PC-YOLOv5s	49.5	37.8	38.7	21.4	42.9	5.427
YOLOv8s	51.1	38.3	39.4	23.5	43.8	11.129
PC-YOLOv8s	54.1	42.2	43.9	26.5	47.4	7.439
YOLO11s	50.2	38.6	39.5	23.6	43.6	9.416
PC-YOLO11s(Ours)	54.3	41.9	43.8	26.3	47.3	7.103

**Table 3 sensors-25-00348-t003:** Comparison of performance between PC-YOLO11s and other mainstream models in the VisDrone2019 datasets.

**Model**	**P/%**	**R/%**	**mAP@0.5/%**	**mAP@0.5:0.95/%**	**F1**	**Parameter/M**
YOLOv5s	46.3	33.4	33.4	18.1	38.8	7.037
YOLOv6s	48.7	35.9	37.0	22.0	41.3	16.299
YOLOv7-tiny	45.5	39.1	36.0	18.7	41.7	6.021
YOLOv8s	51.1	38.3	39.4	23.5	43.8	11.129
YOLOv9s	50.6	39.1	41.0	24.1	44.1	7.170
YOLOv10s	50.6	37.9	39.1	23.5	43.3	8.042
YOLO11s	50.2	38.6	39.5	23.6	43.6	9.416
PC-YOLO11s (Ours)	54.3	41.9	43.8	26.3	47.3	7.103

**Table 4 sensors-25-00348-t004:** Ablation experiments of PC-YOLO11s model.

Model	P234	CSA	P/%	R/%	mAP@0.5/%	mAP@0.5:0.95/%	F1	Parameter/M
YOLO11s	×	×	50.2	38.6	39.5	23.6	43.6	9.416
	√	×	53.0	41.6	42.9	25.7	46.6	7.082
	×	√	51.0	38.4	39.5	23.8	43.8	9.434
	√	√	54.3	41.9	43.8	26.3	47.3	7.103

**Table 5 sensors-25-00348-t005:** Comparison of the effects of different detection layers.

Model	P345	P2345	P34	P234	P/%	R/%	mAP@0.5/%	mAP@0.5:0.95/%	F1	Parameter/M
YOLO11s	√	×	×	×	50.2	38.6	39.5	23.6	43.6	9.416
	×	√	×	×	53.8	42.3	44.1	26.8	47.4	9.226
	×	×	√	×	51.5	37.8	39.1	23.3	43.6	6.891
	×	×	×	√	53.0	41.6	42.9	25.7	46.6	7.082

**Table 6 sensors-25-00348-t006:** Comparison of the effects of adding different attention mechanisms.

Model	P234	CBAM	CA	CSA	P/%	R/%	mAP@0.5/%	mAP@0.5:0.95/%	F1	Parameter/M
YOLO11s	√	×	×	×	53.0	41.6	42.9	25.7	46.6	7.082
	√	√	×	×	53.1	41.7	43.2	26.2	46.7	7.196
	√	×	√	×	53.0	42.3	43.8	26.3	47.0	7.102
	√	×	×	√	54.3	41.9	43.8	26.3	47.3	7.103

**Table 7 sensors-25-00348-t007:** Comparison of the effects of adding attention mechanisms at different positions.

Model	P234	C3k = False	C3k = True	P/%	R/%	mAP@0.5/%	mAP@0.5:0.95/%	F1	Parameter/M
YOLO11s	√	×	×	53.0	41.6	42.9	25.7	46.6	7.082
	√	√	×	54.3	41.9	43.8	26.3	47.3	7.103
	√	√	√	53.2	42.0	43.6	26.2	46.9	7.114

**Table 8 sensors-25-00348-t008:** Comparison of performance between PC-YOLO11s and other mainstream models in the tea bud datasets.

Model	P/%	R/%	mAP@0.5/%	mAP@0.5:0.95/%	F1	Parameter/M
YOLOv5s	85.0	75.3	83.8	60.1	79.9	7.012
YOLOv6s	82.9	74.4	82.5	61.0	78.4	16.297
YOLOv7-tiny	76.5	70.5	78.1	47.9	73.4	6.014
YOLOv8s	85.4	76.7	84.8	65.3	80.8	11.125
YOLOv9s	85.2	76.9	84.9	66.6	80.8	7.167
YOLOv10s	86.9	75.1	83.9	64.7	80.5	8.035
YOLO11s	86.4	75.9	85.3	66.5	80.8	9.413
PC-YOLO11s (Ours)	87.1	77.3	85.7	68.6	81.9	7.101

## Data Availability

Data are contained within the article.

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
