# Peer review of "PC-YOLO11s: A Lightweight and Effective Feature Extraction Method for Small Target Image Detection"

_sensors, 2025, doi:10.3390/s25020348_

Round 1
Reviewer 1 Report
Comments and Suggestions for Authors
The author proposes two improvement measures to optimize the YOLO11 model, including modifications to the network detection layer and the introduction of a new attention mechanism CSA and its integration into the network module. The result model architecture they submitted is called "PC-YOLO11s" and is tested and compared with some object detection methods on two datasets. These two improvements are reasonable and the paper writing is well-organized. Before considering acceptance, the following issues should be clarified or improved:
1、Table 3、4、5、6 is awkward. The authors should redesign this table to better reveal the roles played by the two different improvements. For example, "xxxx" denotes no improvements, "tick xxx" signifies that only one is used, etc.
2、The effectiveness of the proposed four improvements is not convincing enough as they have only been applied on YOLO11s. In fact, a total of five YOLO11 models have been released, from YOLO11n to YOLO11x. More evidence should be provided to show that the two improvements are also applicable to the other models (at least two more) in this series.
Author Response
Thank you very much for your help. Please refer to the Word document for a detailed reply.

Reviewer 2 Report
Comments and Suggestions for Authors
The authors proposed an improved lightweight network (PC-YOLO11s) to realize high-performance small target image detection. The overall results in this manuscript are technically valid and sound, thus I recommend this work for publication. The following minor problems should be addressed.
(1) The definition of small target (such as the ratio of pixel count) should be given.
(2) At the line of 24, ‘Increased’ should be corrected as ‘increased’.
(3) At the line of 84, the full name of CSA should be given.
Author Response

(The authors gave the same response as above.)

Reviewer 3 Report
Comments and Suggestions for Authors
All comments were written in the review file.

Author Response

(The authors gave the same response as above.)

Round 2
Reviewer 3 Report
Comments and Suggestions for Authors
All comments were written in the review file.

Author Response
Comments 1:In expression (2), mAP should be written correctly. This metric should be written as follows.
Response 1:Thank you very much for your reminder. Following your suggestion, I have made the necessary revisions. Please review them. This will further enhance the rigor of our paper. Once again, thank you for your assistance!